# Characterization of *Kazachstania slooffiae*, a Proposed Commensal in the Porcine Gut

**DOI:** 10.3390/jof7020146

**Published:** 2021-02-17

**Authors:** Katie Lynn Summers, Juli Foster Frey, Ann M. Arfken

**Affiliations:** Animal Biosciences and Biotechnology Laboratory, Beltsville Agricultural Research Center, ARS, USDA, Beltsville, MD 20705, USA; juli.frey@usda.gov (J.F.F.); annma31415@aol.com (A.M.A.)

**Keywords:** *Kazachstania slooffiae*, mycobiome, microbiome, pig, porcine, swine, biofilms, *Lactobacillus*, *Enterococcus*

## Abstract

*Kazachstania slooffiae* is a fungus commonly isolated from the gastrointestinal tract and feces of post-weaning pigs. Studies have implicated its ability to positively alter piglet gut health through potential symbioses with beneficial bacteria, including *Lactobacillus* and *Prevotella*, in providing amino acids as an energy source for microbial and piglet growth, and it has been found to be positively correlated with short-chain fatty acids in the piglet gut. However, basic mycological information remains limited, hampering *in vitro* studies. In this study, we characterized the growth parameters, biofilm formation ability, susceptibility to antimicrobials, and genetic relatedness of *K. slooffiae* to other fungal isolates. Optimal fungal growth conditions were determined, no antifungal resistance was found against multiple classes of antifungal drugs (azoles, echinocandins, polyenes, or pyrimidine analogues), and dimorphic growth was observed. *K. slooffiae* produced biofilms that became more complex in the presence of *Lactobacillus acidophilus* supernatant, suggesting positive interactions with this bacterium in the gut, while *Enterococcus faecalis* supernatant decreased density, suggesting an antagonistic interaction. This study characterizes the *in vitro* growth conditions that are optimal for further studies of *K. slooffiae*, which is an important step in defining the role and interactions of *K. slooffiae* in the porcine gut environment.

## 1. Introduction

Weaning is a critical phase in piglet life and is associated with increased stress, reduced growth performance, and predisposition to opportunistic infections such as post-weaning diarrhea [1,2,3]. The ability to identify potential growth promotants in agricultural animals is critical to reduce infections seen during weaning, promote pork production, and reduce farmer loss. Significant alterations in the composition of the bacteriome and mycobiome have been documented during the weaning transition, but specific interactions between the microbiome and the immune system, that result in altered growth, remain to be determined. The microbiome is an integral player in animal health and recently, members of the rare biosphere [4] have been recognized as being critical in altering bacterial–fungal–host interactions, and therefore, animal health and growth. Members of the mycobiome can alter host health, and disruption of members of the mycobiome can result in disease [5,6,7,8,9,10,11,12]. Furthermore, commensal fungi can alter host immunity during normal health as well as modify the severity of some diseases [13,14,15,16,17,18].

One important member of the rare biosphere in swine is the fungus *Kazachstania slooffiae*. *K. slooffiae* has been found to be the predominant post-weaning fungi in the gastrointestinal (GI) tract and feces of piglets. This fungus persists in adult pigs across geography and may play a role in providing nutritional benefits to pigs [19,20,21,22,23,24,25,26]. *K. slooffiae* is a yeast that is closely related to species such as *Candida (Kazachstania) pintolopesii* and *Candida* (*Kazachstania*) *bovina*, and it is a member of the *Kazachstania* (*Arxiozyma*) *telluris* species complex [27]. Kurtzman et al. utilized strains originally isolated from horse and porcine ceca and described *K. slooffiae* as vegetative with spherical to ellipsoidal cells and multilateral budding, resulting in lobate colony growth and lacking true hyphae. In swine, positive inferred interactions have been found between *K. slooffiae* and beneficial gut bacteria including *Prevotella* and *Lactobacillus* [19,20]. Despite these previous studies, little is known about this fungus and its role in modulating the microbial milieu in the gut, and thus, piglet health, growth, and disease resistance.

In this study, we aimed to further document the growth parameters of piglet-derived *K. slooffiae* under different *in vitro* nutrient conditions, assess microscopic growth characteristics and antimicrobial resistance, characterize related strains based on available sequencing data, and utilize biofilm assays to elucidate the ability of *K. slooffiae* to grow as a biofilm in the presence of different potential bacterial antagonists and synergists. This study lays the groundwork for future studies to assess the role of *K. slooffiae* in altering piglet health through fungal–bacterial interactions during the critical weaning transition.

## 2. Materials and Methods

Culture isolation and sequencing of *K. slooffiae*. Fecal samples were collected from post-weaning piglets (age 22–28 days) using sterile weigh basins and then transferred to sterile 50 mL conical tubes. Samples were weighed, normalized to 0.2 g of feces, 2 mL of sterile 1X PBS was added, and samples were homogenized in a biological safety cabinet. Post-homogenization, samples were serially diluted and plated in triplicate on Sabouraud Dextrose Agar (SDA) and Yeast Potato Dextrose Agar (YPD) (BD Difco, Franklin Lakes, NJ, USA). Both agar types were supplemented with 0.1 mg/mL cefoperazone (cef), a broad-spectrum cephalosporin, to reduce bacterial growth on plates (Sigma-Aldrich, St. Louis, MO, USA). Agar plates were incubated under different temperature conditions (37 °C, 20 °C) and with and without 5% CO_2_ supplementation. Colonies that grew were Sanger sequenced utilizing ITS-2 (Internal Transcribed Spacer region 2) primers for identification (ITS3: 5′-GCATCGATGAAGAACGCAGC; ITS4: 5′-TCCTCCGCTTATTGATATGC) [28]. Sequencing results determined that the predominant fungal species was *Kazachstania slooffiae*, which was previously isolated in pigs [19,21,24,25,26].

For genome sequencing, one colony from a YPD + cef agar plate of sequence-confirmed *K. slooffiae* was used to inoculate 4 mL YPD + cef liquid media. The culture was incubated at 37 °C, 200 rpm, with culture volume gradually increased over the next 2 days by adding fresh YPD + cef to a total of 120 mL. On day 3 of incubation, fungal cells were collected in 4 individual 30 mL aliquots by centrifuging at 100× *g* for 5 min. Supernatants were removed, and pellet dry weights were weighed (0.60–0.76 g). Each pellet was homogenized 3x in liquid N_2_ with a Tissue Homogenizer (Omni International, Kennesaw, GA, USA) and kept on dry ice until genomic DNA was isolated using the DNeasy Plant Maxi Kit (Qiagen, Hilden, Germany). DNA was quantified with the Qubit Flex (Invitrogen, Carlsbad, CA, USA) and visualized with a 0.7% agarose gel to check for high-quality, high molecular weight DNA. Whole-genome sequencing of the *K. slooffiae* isolate was performed on the PacBio Sequel II 18M SMRT Cell platform. The resulting long reads were quality checked, and a draft genome was assembled with FALCON ver.0.5 [29] at the University of Maryland Genomics Resource Center.

**Antifungal susceptibility.** Fungal cultures were started from 1 colony in 5 mL of YPD + cef or SDB + cef and grown at 37 °C, 5% CO_2_ for 24 h. Then, these cultures were inoculated into 20 mL of matching media broth for a total of 25 mL of broth culture. These were allowed to grow for an additional 24 h. Upon growth density, a sterile cotton tip swab was dipped into the broth in a biological safety cabinet and rubbed across an agar plate. This swab was repeated 2 more times, and the culture was allowed to air dry in the BSC. Upon drying, the antifungal or antibiotic of interest was added to the middle of the agar plate, and plates were incubated at 37 °C, 5% CO_2_ and results were read at 24, 48, and 72 h. When possible, Minimum Inhibitory Concentration (MIC) Test Strips (MTS) were used to assess MIC (flucytosine, itraconazole, caspofungin, voriconazole) (Liofilchem, Roseto degli Abruzzi, Italy). Susceptibility or resistance was based on the ability of the antimicrobial to create a zone of fungal growth clearance.

**Growth curve.** Five 5 mL starter cultures of YPD + 0.1 mg/mL cef were inoculated with a single colony each of *K. slooffiae*. After 39 h of incubation at 37 °C, shaking at 200 rpm, twenty 20 mL cultures were created from 1 mL of starter culture + 19 mL YPD + cef. Samples were briefly vortexed before removing a 200 µL aliquot and OD_600_ readings were taken every two hours, from T_0_ to 20 h. Serial plating at T_0_ determined the average starting inoculum to be 4.5 × 10^5^ CFU (colony forming units)/mL. *C. albicans* CHN1 growth curves were performed similarly. However, due to the denser and more rapid growth of this species, 0.5 mL of starter culture + 19.5 mL YPD + cef were started. OD_600_ readings were taken every two hours from T_0_ to 11 h. Serial plating at T_0_ determined the average starting inoculum to be 3.5 × 10^6^ CFU/mL. CFU counts for the growth curve were determined through serial dilutions on YPD + cef agar plates and read after 24–36 h (*n* = 3 for each fungus at each time point).

**Amplicon size determination.** Fungal colonies were grown for *K. slooffiae* and *C. albicans*, as described above, and 5 mL cultures were inoculated with a single colony. After 24 h of growth, whole DNA was isolated using the DNeasy PowerSoil Pro Kit (Qiagen). PCR amplification was performed on the DNA isolated with 18S, ITS-1, or ITS-2 primers (0.03 µM) (Table 1) using the following parameters: 95 °C or 3 min; followed by 35 cycles of 98 °C for 30 s, 60 °C for 3 min, 72 °C for 30 s, and a final extension at 72 °C for 5 min. Samples were run on an Agilent Bioanalyzer 2100 and confirmed on a 2% ethidium bromide gel (data not shown).

Microscopy. Fungal cultures were grown as described above, and wet mounts were captured on an Olympus IX73 microscope (Olympus America, Inc., Center Valley, PA, USA) equipped with a Hamamatsu ORCA-R2 digital CCD camera (Hamamtsu Corporation, Bridgewater, NJ, USA) and measured using the Olympus CellSense Dimension (ver.2.1) software.

Phylogenetic tree. The full ITS gene region (ITS1-5.8S-ITS2) in the assembled *K. slooffiae* isolate draft genome was extracted using ITSx version 1.1.2 [32]. Additional *Kazachstania* sp. ITS gene region sequences were downloaded from the National Center for Biotechnology Information (NCBI) RefSeq database [33]. Remaining SSU and LSU regions in the downloaded ITS gene sequences were identified and removed using ITSx version 1.1.2. Combined *Kazachstania* ITS gene region sequences from RefSeq and this study were aligned with MUSCLE [34]. Phylogenetic trees were constructed using maximum likelihood (ML) with 1000 bootstrap replicates under the Tamura–Nei model [35]. Both alignments and tree construction were conducted in MEGA version X [36]. *Naumovozyma* (*Saccharomyces*) *castellii*, a closely related genus, was selected as an outgroup.

Biofilms. A single colony of *K. slooffiae* was inoculated into 5 mL of broth media of interest and grown at 37 °C for 24 h. Then, this culture was inoculated into 20 mL of broth media and grown for another 24 h. Then, 200 µL of *K. slooffiae* was plated into Falcon sterile, tissue culture grade 96-well, flat-bottom plates (Corning, Glendale, AZ), treatments of interest were added, plates were wrapped with parafilm to prevent evaporation, and they were grown without shaking at 37 °C, 5% CO_2_ for 72 h. For each experiment, the culture of *K. slooffiae* was replica plated onto the corresponding agar type to assess the fungal burden in each well and incubated at 37 °C 5% CO_2_ for 48 h prior to reading. As previously published, biofilms were stained and assessed for optical density [37,38]. Briefly, at 72 h, the supernatant from the biofilm plates was gently removed and biofilms were air-dried for 45 min. Following air-drying, 200 µL of 4% paraformaldehyde was added and plates were incubated at room temperature for 45 min. Following incubation, the paraformaldehyde was removed, and plates were gently washed 4x with 350 µL of sterile 1X phosphate buffered saline (PBS). Then, 110 µL of 4% aqueous crystal violet was added to each well and incubated for 45 min prior to washing 3x with 350 µL sterile water. Next, 200 µL of 95% ethanol was added to each well to destain the biofilms for 45 min. Finally, 100 µL was transferred from each well into a fresh 96-well plate and read at 595 nm on a SpectraMax 340 plate reader (Molecular Devices, San Jose, CA, USA). Supernatants from bacterial isolates were used to test alterations on *K. slooffiae* biofilm density: *Enterococcus faecalis* ATCC BAA-2128 (piglet feces isolate), *Lactobacillus acidophilus* ATCC 53671 (swine intestinal isolate), and *Lactobacillus fermentum* ATCC 23271 (human intestinal isolate). Bacterial cultures were grown from glycerol stock stored at −80 °C by inoculating into 5 mL of De Man, Rogosa, Sharpe broth (MRS) and incubated at 37 °C for 24 h without shaking. Following bacterial growth, cultures were spun down for 5 min at 211× *g*, supernatant was removed, and 50 µL of supernatant was added to each biofilm well. Replica plating of bacterial cultures were performed on MRS agar plates to confirm bacterial numbers.

Statistics. Statistical analyses were performed in GraphPad Prism version 8.4.3. (GraphPad Software, San Diego, CA, USA). The statistical significances between biofilm growth were determined using a one-way ANOVA with a Tukey post-test. Values of *p* < 0.05 were considered significant.

## 3. Results

Growth of *K. slooffiae. K. slooffiae* was isolated from fecal samples directly sampled from piglets in the week post-weaning (age 22–28 days) and grown as described in the methods. Confirmed isolates were grown for 24–72 h on Yeast Potato Dextrose (YPD) and Sabouraud Dextrose (SDA) nutrient agars (Figure 1). Colonies were shiny, cream-colored with lobate margins. While colonies grown on YPD displayed a concave elevation (Figure 1A,B), colonies grown on SDA displayed a flat elevation (Figure 1C,D). YPD broth culture grown at 37 °C without shaking both displayed sediment growth (Figure 1E,F). Sediment growth was also seen in YPD broth at 37 °C with shaking at 200 rpm and in cultures grown in SDB with or without shaking (not shown). As seen in previous literature, *K. slooffiae* grows as a spherical, yeast, or as pseudohyphae (Figure 2), and budding yeasts were seen (Figure 2B). Pseudohyphae were scarcely differentiated, and no ascopores were seen in our experiments.

To assess the growth rate of *K. slooffiae* in comparison to another gut fungi, we performed growth curves for *K. slooffiae* and *Candida albicans* CHN1 (human gut isolate). Cultures were followed for 26 h or longer, and growth for *C. albicans* CHN1 outgrew *K. slooffiae* regardless of nutrient broth. Exponential growth for *K. slooffiae* was seen between 1 and 8 h with a plateau at OD_595nm_ 0.843 (Figure 3). Colony counts were assessed throughout the time course, and at 4 h of growth, *K. slooffiae* had a density of 9.50 × 10^6^ CFU, which appeared to plateau through 26 h of growth, where the CFU remained similar at 9.58 × 10^6^. *C. albicans* CHN1 cultures demonstrated faster growth with 4.11 × 10^7^ CFU by 4 h, plateaued at 8 h with 1.91 × 10^8^ CFU, and 1.61 × 10^8^ CFU at 26 h of growth. *K. slooffiae* was assessed for antimicrobial resistance to four classes of antifungal drugs and two antibiotics as controls. *K. slooffiae* was grown on YPD agar and SDA, and growth inhibition was assessed through Minimum Inhibitory Concentrations (MIC). No difference in antimicrobial susceptibility was seen between the two growth conditions (SDA not shown), and no resistance was seen for azoles (fluconazole, voriconazole, itraconazole), echinocandins (caspofungin), polyenes (amphotericin B), pyrimidine analogues (flucytosine) or antibiotics (ampicillin (beta-lactam), cefoperazone (cephalosporin)) (Table 2).

Molecular studies. Whole DNA was isolated from *K. slooffiae* and *C. albicans* CHN1 and PCR amplification was performed for 18S, ITS-1, and ITS-2 fungal-specific primers sets that were previously published (Table 1). PCR amplicons were assessed by Agilent Bioanalyzer (Figure 4A) and confirmed on ethidium bromide gel (data not shown). Results are summarized to demonstrate expected amplicon size based on primer selection (Figure 4B). A total of *n* = 16 full length ITS gene copies (ITS1-5.8S-ITS2) were extracted from the draft *K. slooffiae* genome and were confirmed as *K. slooffiae*, a member of the *Kazachstania telluris* complex, based on phylogenetic analysis (Figure 5). One gene identified with a length of 13,219 bp was removed from analysis due to being greater than the predicted size of the ITS gene region. Among these copies, *n* = 7 non-identical ITS genes were detected at >98.8% similarity, with lengths ranging from 719 to 721 bp and divergence ranging from 1 to 8 nucleotides.

Biofilm growth. Next, we assessed the ability of *K. slooffiae* to grow in a biofilm. While multiple medias and time points were assessed, 72 h and YPD + cef were found to be the optimum conditions for promoting *K. slooffiae* biofilms (data not shown). *K. slooffiae* was grown in YPD + cef broth, and serial dilutions of culture were added into each well of a 96-well tissue culture grade plate, wrapped with parafilm to prevent evaporation, and grown for 72 h and then assessed for biofilm formation by optical density reading at 595 nm (Figure 6). While biofilm growth showed a dilution trend at lower dilutions, starting culture above 10^4^ did not continue to become more optically dense, suggesting no further biofilm complexity. Bacterial supernatants from swine and human isolates were tested to examine if they could alter *K. slooffiae* biofilm complexity. *Lactobacillus fermentum* (human intestinal isolate) supernatant did not significantly alter *K. slooffiae* biofilms, while *Lactobacillus acidophilus* (swine intestinal isolate) supernatant significantly increased the optical density of the biofilms, suggesting an increase in biofilm complexity (Figure 6). Interestingly, the supernatant from a known opportunistic pathogen, *Enterococcus faecalis* (swine feces isolate) significantly inhibited *K. slooffiae* biofilm density, suggesting a potential antagonism.

## 4. Discussion

Due to the potential importance of *K. slooffiae* in piglet health, we aimed to investigate optimal growth conditions, biofilm formation ability, expected DNA amplicons, and other in vitro assessments. *K. slooffiae* was able to efficiently grow on YPD or SD agars but displayed colony morphology alterations based on the nutrient base (Figure 1). Other fungal species have demonstrated convex or concave colony morphology patterns based on glucose availability [39], so we hypothesize that the morphology differences are due to sugar availability variances between the two agar types. Broth cultures of *K. slooffiae* displayed sedimentary growth regardless of incubation with or without shaking or growth in YPD or SDB (Figure 1E,F, data not shown).

Microscopy confirmed fungal growth as seen in previous work. *K. slooffiae* grows as a yeast and can produce pseudohyphae (Figure 2). A study by Kurtzman et al. demonstrated a single isolate of *K. slooffiae* that was able to form ascopores; however, no ascopores were seen in our isolate from piglet feces under any conditions [27]. It should be noted that the strain in the Kurtzman study required 15–20 days of growth at 25 °C, and while this did not occur with our piglet-derived strain, it cannot be ruled out that this strain could potentially create ascopores under certain circumstances. Ascopores are spores formed within an ascus, a cylindrical-shaped sac, and they are specifically made by members of the *Ascomycota*. As with other types of spores, ascopores are utilized for survival and dispersion [40], and other *Kazachstania* species have been found to form ascopores [41]. Ascopores formed by other fungal species have demonstrated their contribution to virulence, especially in causing diseases of plants [42,43,44], but they have also been found to assist in survival through the gastrointestinal tract of *Drosophila* [45]. Other microbes rely on sporulation to survive gut conditions [46]; therefore, ascopore formation cannot be ruled out as an important factor for *K. slooffiae*. The role of ascopores in piglet gut survival for *K. slooffiae* is unknown, but future studies must be done to ascertain how many strains of *K. slooffiae* form ascopores and under which conditions, as they may play an important role in the survival and transmission in piglets.

In order to assess the growth of *K. slooffiae,* we performed growth curve analyses. For comparison, we concurrently analyzed the human gut fungal isolate, *C. albicans* CHN1. Similar to *K. slooffiae*, *C. albicans* is a polymorphic yeast found in healthy GI tracts and feces [47], but while *C. albicans* is a commensal, it is also a well-documented opportunistic pathogen. We propose that *K. slooffiae* is a gut commensal in pigs, but currently, little is known about the potential virulence of *K. slooffiae.* Our growth curves demonstrate that under laboratory conditions, *C. albicans* is able to grow to a significantly higher density in YPD or SDB than *K. slooffiae* (Figure 3 and data not shown). While this rapid fungal growth may be specific to the laboratory setting, it could also demonstrate the ability of *C. albicans* to grow rapidly in the GI setting, promoting its commensalism through interactions with the gut microbiome and host immune response. However, *C. albicans* could also utilize this rapid growth to assist its opportunistic pathogenicity. *K. slooffiae* grows slower under all tested laboratory conditions, but the implications of this comparatively reduced growth in vivo remain to be assessed. In vivo growth conditions may be more optimal to promote faster growth for *K. slooffiae*, or perhaps its slow growth is retained in vivo and plays a role in immune recognition not resulting in clearance. Future work necessitates the sequencing of the *K. slooffiae* genome to provide genetic comparisons to elucidate any potential virulence factors utilized by fungal pathogens, such as adhesins, dimorphism, phospholipases, and aspartyl proteases [48,49].

Fungal identification alone is not sufficient when investigating complex environments such as the porcine gut. Further, fungal-specific primers have been documented to miss certain species entirely. To assess the ability of fungal-specific primers to amplify the DNA from *K. slooffiae*, we isolated whole DNA and performed PCR amplification with three different previously published primer sets (Figure 4A). Primers for 18S, ITS-1, and ITS-2 were all able to efficiently amplify these target sequences in *K. slooffiae*, and the resulting amplicon sizes are documented here (Figure 4B). A caveat of these results is that the DNA was isolated from a homogenous culture and not from heterogenous sample such as feces or luminal contents, simplifying fungal amplification. Despite the simplicity of the amplification, these data are important when assessing *K. slooffiae* in the context of the piglet gut and feces, as different fungal primer sets can have different efficiencies based on the environmental sample being assessed. These data establish accurate expected amplicons to help identify this fungus in the future. We have previously published work suggesting that ITS-2 specific primers are the appropriate choice for piglet gut and fecal samples [21], but we also suggest a combinatorial approach when studying a new sample set, utilizing more than one fungal-specific primer set [50].

In addition to targeted amplification, we also examined the complete ITS gene region from our *K. slooffiae* isolate and then extracted an assembled draft genome of our isolate based on whole genome shotgun sequencing. This allowed for a full-length ITS phylogenetic identification and comparison of *K. slooffiae*, as well as the determination of potential repetitive ITS regions and heterozygosity. Since there is currently no available published genome of *K. slooffiae* and the heterozygosity and polyploidism of this species is unknown, a complete assembly of this genome remains a challenge. Based on our draft assembly, 16 copies of the ITS gene were detected with seven different copy variations and were identified as *K. slooffiae* within the *Kazachstania telluris* complex. While some of these copies may be a result of potential sequencing errors and unresolved consensus sequences, ITS repeats in the genome may also be due to multiple copies of rDNA present in many fungal species. In fungi, rDNA copies are estimated to range from 14 to 1422 copies [51] and may vary due to environmental conditions [52]. Furthermore, the variation found in the *K. slooffiae* ITS copies may result from heterozygosity or polyploidism, which is known to be prevalent in many types of fungi [53,54]. Additional analysis including the qPCR of rDNA copy numbers and newer sequencing technology such as Hi-Fi [55] will provide further insight into the *K. slooffiae* genome.

While fungi have recently gained interest in the medical field due to their drug resistance, such as *Candida auris*, our studies suggest that *K. slooffiae* does not currently possess significant drug resistance (Table 2) [56]. Our current data do not suggest that *K. slooffiae* behaves as an opportunistic pathogen, but that cannot be ruled out. Resistance to polyenes, pyrimidine analogues, azoles, and echinocandins have been seen in similar fungi, such as *C. albicans*, and these resistances were often due to simple mutations in one to two genes [57]. *C. auris* has been found to be highly drug resistant, which is a critical health concern due to a lack in novel antifungal drugs [58]. Recently, fluconazole resistance was seen in uncommon yeast pathogens, including *Kazachstania telluris, Kazachstania bovina, Kazachstania exigua,* and *Kazachstania servazzii* [59]. The ability of *K. slooffiae* to become drug resistant remains to be elucidated, but due to the rarity of *Kazachstania* sp. becoming antifungal or multi-drug resistant, it seems unlikely that resistance will become a problem.

We performed preliminary experiments to assess the ability of *K. slooffiae* to form biofilms. Similar to the susceptibility testing of fungi to antifungals, experiments were carried out to identify major inhibitory effects. Utilizing previously described biofilm protocols, we assessed growth conditions including time of growth, fixation method utilized, and nutrient conditions used [37,38]. After optimizing conditions, we assessed a dilution series of *K. slooffiae* to determine if the starting culture density affected biofilm development. Unexpectedly, we found that higher density starting cultures did not create more optically dense biofilms. In fact, a lesser range of 10^2^-10^4^ was found to be optimal when starting biofilms, suggesting quorum sensing involvement in optimal biofilm growth (data not shown). Next, we assessed the ability of gut bacterial supernatants to alter *K. slooffiae* biofilm formation. Previous studies have found the ability of bacterial supernatants to alter bacterial and fungal biofilm growth [60,61,62], and our inferred interactions suggest this may be occurring in the piglet gut. We found that while *L. acidophilus* supernatant was able to enhance biofilm formation, *L. fermentum* did not significantly alter biofilm density. Supernatant from a strain of *E. faecalis* isolated from piglet feces significantly decreased the optical density of *K. slooffiae* biofilms, suggesting an antagonism. While the composition of the supernatant remains to be determined, cross-species quorum-sensing molecules or secreted molecules such as capsules could play a role [63,64]. Future studies will assess the biofilm composition through microscopy and will determine the composition of the bacterial and fungal supernatants for quorum sensing molecules or other molecules of interest. Studies will be done to assess the role of supernatant acidities and the effect of pH on biofilm density and formation. One other factor of interest is the role of ethanol production as a bacterial product that plays a role in impeding biofilm growth and will need to be investigated further [65]. Furthermore, differences in supernatant effects on planktonic versus biofilm *K. slooffiae* are of interest to determine the quorum-sensing molecules utilized by this fungus.

The weaning transition is a critical period in a piglet’s life when piglets are removed from their mothers, placed into new housing pens, and started on solid diet. These significant changes are associated with a dramatic shift in microbial communities, including the mycobiome. Previous studies have demonstrated conflicting data on which porcine gut organs harbor the most *K. slooffiae* [19,21,25,64], but overall, *K. slooffiae* has been found to become the predominant fungal species found in the feces and gut organs of healthy, post-weaning pigs. However, details on its growth or interactions with the bacteriome are lacking. Previous inferred interactions have suggested that *K. slooffiae* has positive interactions with beneficial bacteria such as *Prevotella* [19,20]. *K. slooffiae* was previously assessed for its ability to utilize organic acids and amino acids as carbon and energy sources [22], and this fungus was not able to utilize lactic, butyric, proprionic, or acetic acid as a sole carbon source *in vitro*; however, this ability in the gut environment cannot be ruled out [22]. Further, Urubschurov et al. found that *K. slooffiae* does not use free lysine as a sole nitrogen source, eliminating this as a potential competition with the host. These authors hypothesized that one way that *K. slooffiae* could benefit the host was through the ability to form dehydroascorbic acid, which could be beneficial for animals. *K. slooffiae* is able to utilize glucose but has not been documented as utilizing galactose [22,26,27]. They hypothesized that *K. slooffiae* may be a protein source to piglets. This underscores the fact that pre-weanling piglets, that only ingest milk, do not have significant *K. slooffiae* populations, which is perhaps due to an inability to utilize the galactose in lactose. Glucose consumption is also seen in the bacterial species tested in biofilms, and while *L. acidophilus* continues to produce glucose over multiple days, glucose production by *L. fermentum* is reduced after 48 h, which may be a factor limiting fungal growth in the mixed biofilm [65].

Due to the potential for important interactions in the porcine GI tract, more work must be done to determine bacterial–fungal–host interactions that can alter piglet growth performance. One previous study fed *K. slooffiae* to piglets approximately one-week post-weaning, and no direct growth performance changes were seen [22] but gut colonization was seen with only one feeding supplementation. Despite this sole study, inferred interactions suggest that beneficial bacteria and *K. slooffiae* may be having positive interactions. Positive biofilm interactions between *K. slooffiae* and *L. acidophilus* support the potential for interactions in the gut setting. Equally interesting are the negative interactions seen in the biofilm studies involving *E. faecalis*; investigations elucidating the potential for microbial inhibition in the gut setting remain of interest. Furthermore, supplementation leading up to weaning has not been assessed in vivo and may help in preventing post-weaning opportunistic diseases and piglet weight loss. Previous studies have found that many of the fungal species found in the gut were transient due to their nutritional and environmental needs. It is possible that much of the piglet mycobiome is made up of transient fungi introduced in the feed and other environmental factors. However, the mycobiome is also able to alter health and is more easily altered than the bacteriome, creating the potential for an effective piglet health intervention strategy. In this study, we provided comprehensive growth parameters for *K. slooffiae in vitro* and the effects of gut and fecal bacterial samples on biofilm development and density. Further explorations in the piglet and laboratory setting will provide a greater understanding of the complex interactions occurring in the piglet gut milieu that can lead to alterations in piglet growth and health.

## Figures and Tables

**Figure 1 jof-07-00146-f001:**
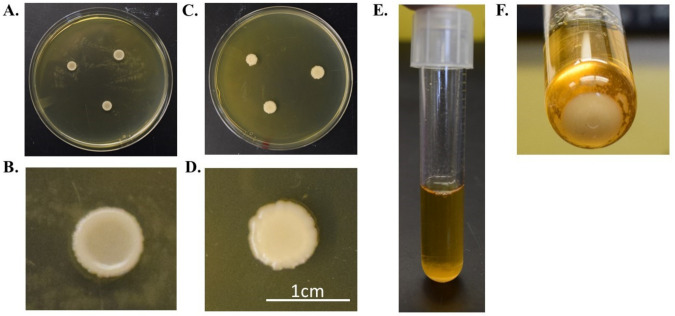
Morphology of *Kazachstania slooffiae*. *K. slooffiae* was grown for 24 h on Yeast Potato Dextrose Agar (YPD) (**A**) and Sabouraud Dextrose (SD) (**C**) agars. Individual colony morphology on YPD demonstrates concave growth (**B**) and SD demonstrated flat colony growth (**D**). Growth in 5 mL YPD broth (**E**,**F**).

**Figure 2 jof-07-00146-f002:**
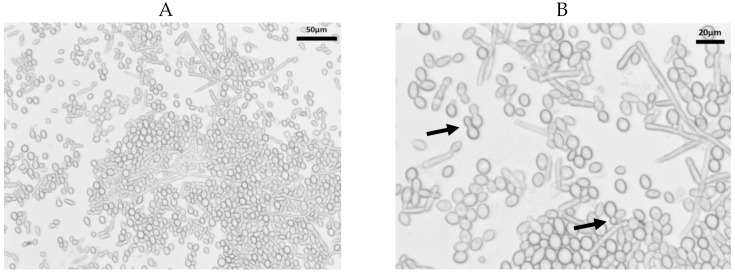
Microscopy of *Kazachstania slooffiae*. Polymorphic growth of *K. slooffiae*. *K. slooffiae* was grown in YPD + cef broth for 72 h and visualized by wet mount microscopy. (**A**) 20× objective displaying yeast and hyphal growth. (**B**) A mixed culture with multiple pseudohyphae seen and budding yeasts (arrows).

**Figure 3 jof-07-00146-f003:**
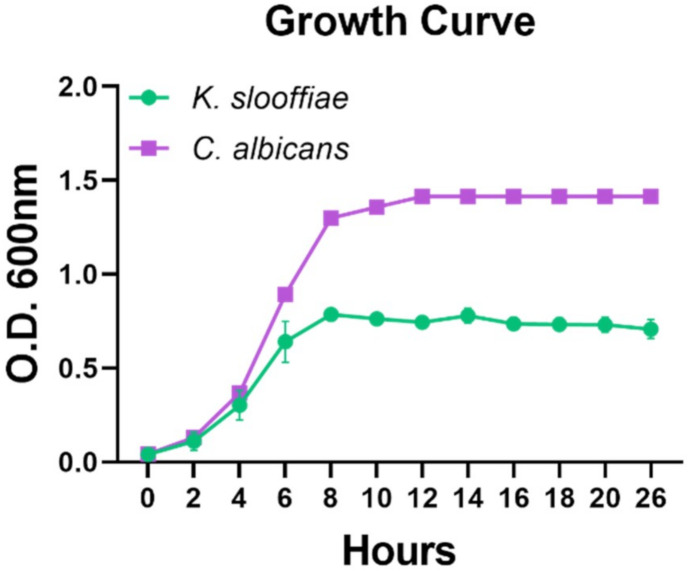
**Growth curves of planktonic fungi.** A) *Kazachstania slooffiae* (pig isolate, *n* = 26 replicates) and *Candida albicans* CHN1 (human isolate, *n* = 9 replicates). Data shown are the mean of replicates with error bars representing the standard deviation.

**Figure 4 jof-07-00146-f004:**
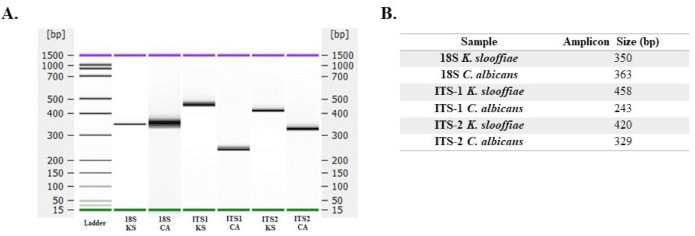
DNA amplicon size based on fungal-specific primer target. DNA isolated from *K. slooffiae* (KS) or *C. albicans* (CA) was amplified utilizing previously published primers (Table 1) for 18S, ITS-1, and ITS-2. Samples were run on an Agilent Bioanalyzer 2100; resulting amplicon sizes are shown by gel (**A**), and expected amplicon sizes are summarized (**B**).

**Figure 5 jof-07-00146-f005:**
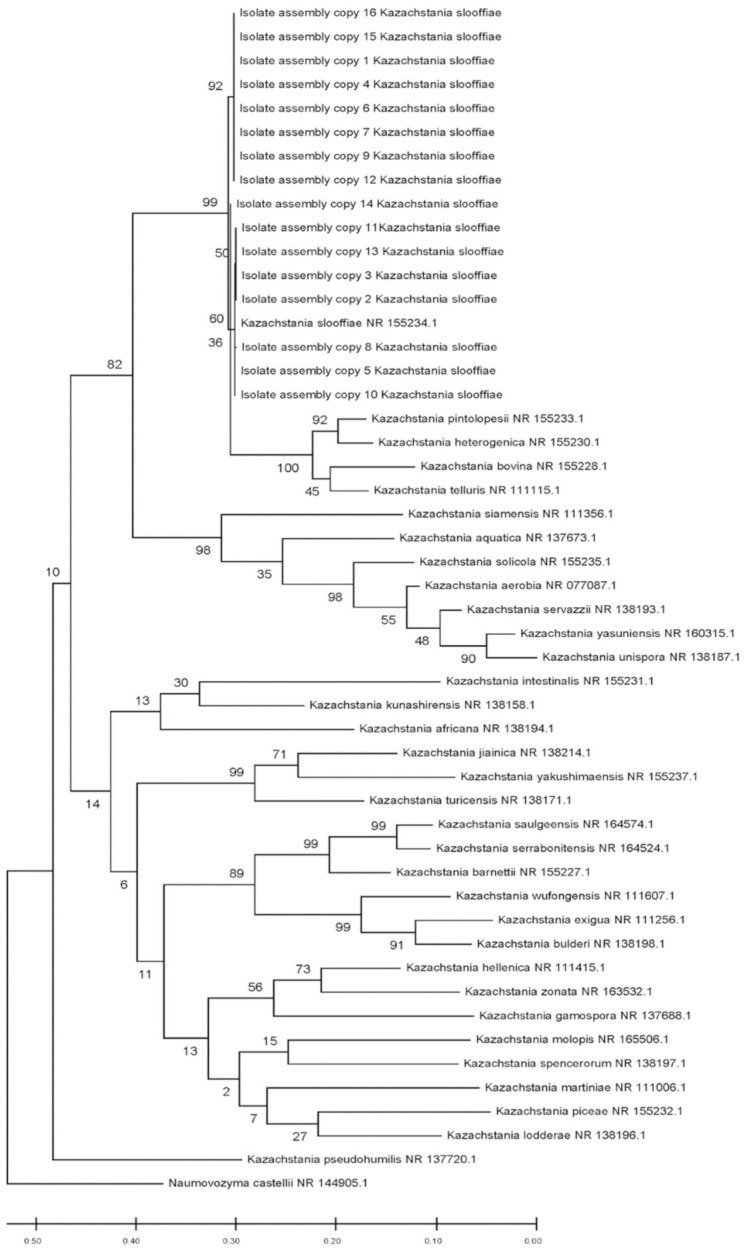
Phylogenetic tree of ITS gene (ITS1-5.8S-ITS2) sequences from *Kazachstania* sp. Accession numbers of reference species downloaded from the RefSeq database indicated at right of species name. *K. slooffiae* genes identified in this study indicated as ‘Isolate assembly copy’. Maximum likelihood tree constructed using Tamura-Nei model and 1000 bootstrap replicates. Bootstrap values located above tree branches. Branch lengths measured in the number of substitutions per site.

**Figure 6 jof-07-00146-f006:**
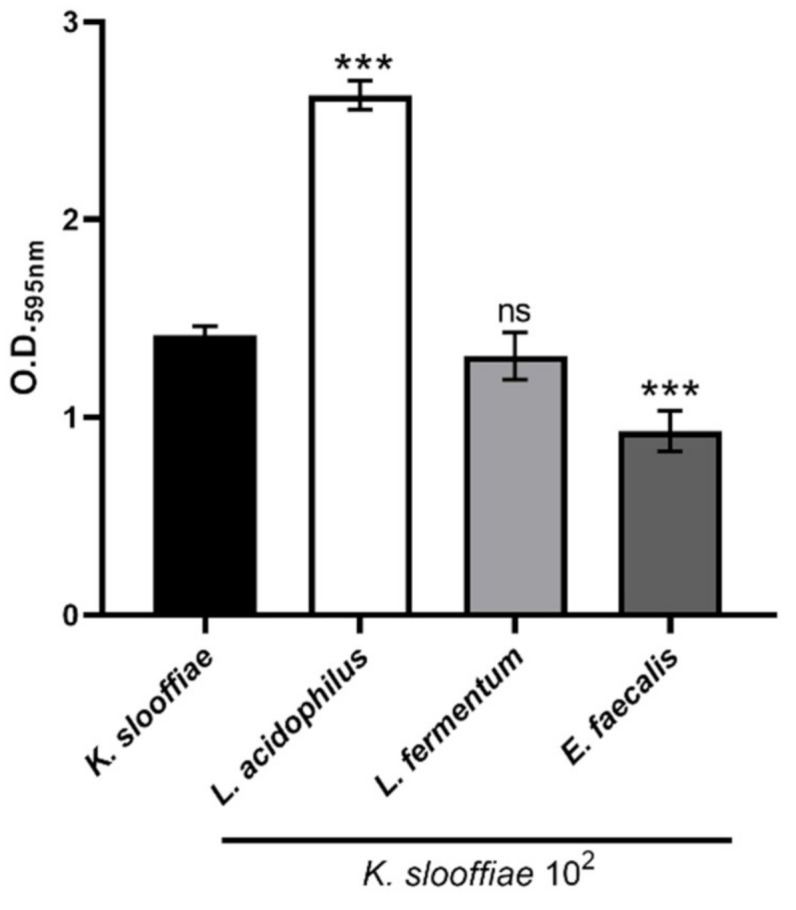
**Biofilm assessment of *K. slooffiae***. *K. slooffiae* was grown in De Man, Rogosa, Sharpe broth (MRS) broth for 72 h without shaking at 37 °C and 5% CO_2_. Following fixation, biofilms were stained with 0.4% crystal violet, washed, and O.D. readings were taken at 595 nm. Bacterial supernatants were co-incubated with *K. slooffiae* biofilms to assess any alterations in the presence of *Lactobacillus acidophilus*, *Lactobacillus fermentum*, and *Enterococcus faecalis*. Data shown is three combined experiments of seven total experiments. (*n* = 16 to 64). One-way ANOVA with a Tukey post-test was performed to determine significance. Error bars represent the standard deviation and *** represents *p* < 0.0001 in comparison to *K. slooffiae* alone.

**Table 1 jof-07-00146-t001:** Fungal-specific primers.

Primer Target	Primer Name	Primer Sequence (5′ -> 3′)	Citation
**ITS-1 Forward**	ITS1	CTTGGTCATTTAGAGGAAGTAA	[30]
**ITS-1 Reverse**	ITS2	GCTGCGTTCTTCATCGATGC	[30]
**ITS-2 Forward**	ITS3	GCATCGATGAAGAACGCAGC	[28]
**ITS-2 Reverse**	ITS4	TCCTCCGCTTATTGATATGC	[28]
**18S Forward**	FF290F	CGATAACGAACGAGACCT	[31]
**18S Reverse**	FR-1R	ANCCATTCAATCGGTANT	[31]

**Table 2 jof-07-00146-t002:** Antifungal susceptibility.

Antimicrobial	Susceptible [Y/N]	Antimicrobial Concentration	Drug class
**Cefoperazone**	N	100 mg/mL	3^rd^ generation cephalosporin
**Ampicillin**	N	100 mg/mL	Beta lactam (penicillin)
**Caspofungin**	Y	0.094 ug/mL	Echinocandins
**Itraconazole**	Y	0.25 ug/mL	Azoles
**Voriconazole**	Y	0.023 ug/mL	Azoles
**Fluconazole**	Y	20 ug/mL	Azoles
**Amphotericin B**	Y	250 mg/mL	Polyenes
**Flucytosine**	Y	1.0 ug/mL	Pyrimidine analogue

## Data Availability

The datasets generated for this study can be found using the Accession number SUB9028074, https://submit.ncbi.nlm.nih.gov/subs/genbank/SUB9028074/ (accessed on 16 February 2021).

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
