# Peer review of "Characterization of Kazachstania slooffiae, a Proposed Commensal in the Porcine Gut"

_jof, 2021, doi:10.3390/jof7020146_

Round 1

Reviewer 1 Report

In general lines, It is difficult to know how many different strains were included in the study and in any test carried out. For example, in figure 3, I guess 26 replicates from the same strain, buy maybe more strains could be included.

Line 97: Antifungal tested?

Line 99: In growth curve, have the authors any plate count? It should be interesting to carry out this point to compare the growth.

Table 2: It should be located in material and methods section.

Figure 6: It should be more interesting to graph the optical density value instead of percentage based on K. slooffiae and also add in this case standard deviation in the figure.

The authors did whole genome sequencing of the strain but there is no any information of genes found, virulence factors as well as other information related to the genome in the results section.

Author Response

Dear Reviewer 1,

Thank you for taking the time to read and assess our manuscript. We are grateful for your helpful comments and critiques. We provide our response below:

In general lines, It is difficult to know how many different strains were included in the study and in any test carried out. For example, in figure 3, I guess 26 replicates from the same strain, buy maybe more strains could be included. Thank you for the request for clarification. This study is investigating a single strain found from sampling piglets at the Beltsville Animal Research Center. We do not have access to a bank of strains at this time. We believe that isolating and characterizing additional strains are beyond the scope of this preliminary characterization. 

Line 97: Antifungal tested?  We have clarified this line to include the 4 antifungal drugs to which MIC strips were available: flucytosine, itraconazole, caspofungin, and voriconazole.

Line 99: In growth curve, have the authors any plate count? It should be interesting to carry out this point to compare the growth. At your suggestion, we have repeated our growth curve studies to include corresponding plate counts of the fungal species at repeated time points. They are now included in the manuscript in the results and materials/methods.

Table 2: It should be located in material and methods section. We agree and have moved table 2 to the material and methods section. It is now table 1 and we have changed the corresponding information throughout the document.

Figure 6: It should be more interesting to graph the optical density value instead of percentage based on K. slooffiae and also add in this case standard deviation in the figure. We have made these corrections, uploaded a new figure and changed the corresponding text and legend to reflect these changes.

The authors did whole genome sequencing of the strain but there is no any information of genes found, virulence factors as well as other information related to the genome in the results section. Thank you for this suggestion. We are very excited about this genome being sequenced. This is the first reporting of this fungal genome being ever sequenced. As stated we utilized a PacBio sequencer and afterwards we have attempted 6 different assemblies utilizing the programs, Falcon and Canoe, under the guidance of genomic experts. Unfortunately, this genome appears to be highly heterozygous and complex. This genome assembly will be a substantial undertaking and we have been told it will take an expert genomic biologist over a year to assemble. We are working toward this exciting goal in our laboratory as we agree that this will be vital information to discuss. However, due to this complexity of this genome and a lack of an accurate assembly, no annotation can be done at this stage and therefore we cannot include any discovered virulence factors in the discussion. 

Thank you for your time and consideration.

Reviewer 2 Report

The gastrointestinal tract's microbiome consists of bacteria, viruses, archea, fungi and yeasts, and protozoa. However, the attention has been paid mainly to bacteriome, but other members of the complex GIT microbiome has been usually forgotten. Summers et al. paid their attention to a fungus Kazachstania sloofiae as a part of the GIT microbiome. Their study was performed in vitro, but the authors discussed both in vitro and in vivo aspects of fungi as a member of the GIT microbiome.

The manuscript is interesting and readable written. However, I have several notices and recommendations for the manuscript.

L60-157: I miss a description of any statistical evaluation in the Material and Methods. However, in Results on line 252 in the figure legend, you wrote: One-way ANOVA with a Tukey post-test was ... Please, describe used statistic evaluation in Material and Methods too.

L76: It would be suitable to add that 200 rpm was shaking as you did on line 166.

L78: 100xg is very low force. Are you sure about this value?

L113: uM - please, use a Greek micro (symbol). This incorrectness dealing with micro is repeated throughout the manuscript. Please, find and correct it in all cases. I will not notice it more.

L113: 95C - please add a degree.

L155: 1000 rpm - What does it mean 1000 rpm? I don´t know the radius of the used rotor. Thus, I am not able to imagine the used force. It is necessary to ad the radius of the rotor or express the conditions using centrifugation force in x g as you did it in other cases.

L195-196: Which characteristics are depicted in Figure 3 (mean, SD, SEM)? It should be written in the figure legend. No statistically significant differences were found? Did you not find any variability in the case of C. albicans (SD, SEM, …)?

L247-253. You performed a statistical comparison by a parametric test. You probably depicted means. However, I miss SD or SEM and also a description of the expressed characteristics in the figure 6 legend.

L261: Agarose or agar?

L306: The primers were not able to amplify K. sloofiae, but a specific nucleotide sequences of the used K. slooffiae-derived DNA template. Please, modify the text.

Author Response

Dear Reviewer 2,

Thank you for taking the time to read and critique our manuscript. We appreciate the efforts you took in assessing our submission. We are submitting our response to each of your comments below:

L60-157: I miss a description of any statistical evaluation in the Material and Methods. However, in Results on line 252 in the figure legend, you wrote: One-way ANOVA with a Tukey post-test was ... Please, describe used statistic evaluation in Material and Methods too. Thank you for this suggestion. We have now included a "Statistics" section to our M/M section and included the statistical analyses performed and with which programs. 

L76: It would be suitable to add that 200 rpm was shaking as you did on line 166. We have included this clarification at L76 to include "shaking at 200rpm".

L78: 100xg is very low force. Are you sure about this value? This is the correct value. We have found that gentle force is preferred by this species, and so a gentle spin was used prior to these growth curves.

L113: uM - please, use a Greek micro (symbol). This incorrectness dealing with micro is repeated throughout the manuscript. Please, find and correct it in all cases. I will not notice it more. Thank you for catching this oversight. We have changed uM and uL to the Greek micro symbol throughout the manuscript. 

L113: 95C - please add a degree. This has been added. 

L155: 1000 rpm - What does it mean 1000 rpm? I don´t know the radius of the used rotor. Thus, I am not able to imagine the used force. It is necessary to ad the radius of the rotor or express the conditions using centrifugation force in x g as you did it in other cases. We have changed rpm to xg in the manuscript to reflect the specific force used.

L195-196: Which characteristics are depicted in Figure 3 (mean, SD, SEM)? It should be written in the figure legend. No statistically significant differences were found? Did you not find any variability in the case of C. albicans (SD, SEM, …)? We have altered figure 3 to include an updated figure and corresponding figure legend to describe the data as the mean with error bars representing the standard deviation. While the K. slooffiae demonstrates a small amount of variability in O.D. readings, the C. albicans did not display great variation and therefore the error bars are not discernable. Please see new figure and legend.

L247-253. You performed a statistical comparison by a parametric test. You probably depicted means. However, I miss SD or SEM and also a description of the expressed characteristics in the figure 6 legend. Based upon your comment and the comments of other reviewers, we have changed figure 6 to now show the O.D. values and the SEM to more clearly show variation in biofilm density. The statistics have been added to the new "Statistics" section in the M/M section.

L261: Agarose or agar? We have corrected this to state "agar" here and at two other locations in the manuscript. 

L306: The primers were not able to amplify K. sloofiae, but a specific nucleotide sequences of the used K. slooffiae-derived DNA template. Please, modify the text. We have modified the text.

Thank you for your time and consideration.

Reviewer 3 Report

Material and Methods – the materials (for example agar, antimicrobial) used in this study must be specific more detail (at least the Producer, City and Country).

Material and Methods – The information about statistical programme and methods used for statistical processing of results is missing.

Author Response

Dear Reviewer 3,

Thank you for taking the time to read and critique our manuscript. We appreciate the efforts you took in assessing our submission. We are submitting our response to each of your comments below:

Material and Methods – the materials (for example agar, antimicrobial) used in this study must be specific more detail (at least the Producer, City and Country). Thank you for catching this oversight. It has now been added to the manuscript for agar, antibiotics, and antifungals.

Material and Methods – The information about statistical programme and methods used for statistical processing of results is missing. Thank you for your suggestion. We have included a statistical section in this revised manuscript.

Thank you for your time and consideration. 

Round 2

Reviewer 1 Report

The authors have corrected the manuscript according the reviewer recommendations. 

Author Response

Thank you so much for your time and efforts.

Dr. Summers

Reviewer 2 Report

Dear Authors,

Many thanks for your acceptance of my recommendations. However, it is necessary to add a used statistical test to the figure 3 legend as it was done in figure legend 6.

Author Response

Dear Reviewer 2,

Could you please clarify which statistical test you would like performed for figure 3? We stated the line represents the mean of the growth curve and the errors bars are standard deviation. Would you like us to run statistical analyses in addition to this between the two fungi in addition? We just want a little clarification on the request.

Thanks,

Dr. Summers